# Quick model-based viscoelastic clot strength predictions from blood protein concentrations for cybermedical coagulation control

Damon E. Ghetmiri[1,7], Alessia J. Venturi [1], Mitchell J. Cohen[2,3] & Amor A. Menezes [1,4,5,6]

Cybermedical systems that regulate patient clotting in real time with personalized blood product delivery will improve treatment outcomes. These systems will harness popular viscoelastic assays of clot strength such as thromboelastography (TEG), which help evaluate coagulation status in numerous conditions: major surgery (e.g., heart, vascular, hip fracture, and trauma); liver cirrhosis and transplants; COVID-19; ICU stays; sepsis; obstetrics; diabetes; and coagulopathies like hemophilia. But these measurements are time-consuming, and thus impractical for urgent care and automated coagulation control. Because protein concentrations in a blood sample can be measured in about five minutes, we develop personalized, phenomenological, quick, control-oriented models that predict TEG curve outputs from input blood protein concentrations, to facilitate treatment decisions based on TEG curves. Here, we accurately predict, experimentally validate, and mechanistically justify curves and parameters for common TEG assays (Functional Fibrinogen, Citrated Native, Platelet Mapping, and Rapid TEG), and verify results with trauma patient clotting data.

Trauma remains the leading cause of death between the ages of 1 and 44[1], with the majority of deaths and essentially all morbidity driven by coagulation and inflammation biology[2]. While much is known about mechanistic drivers, clinical phenotypes, and biologic endotypes of the post-trauma milieu, real-time hemostatic state identification, future coagulation trajectory prediction, and overall therapeutic clinical decision-making are hindered by inadequate models with slow-arriving inputs. Smart devices are desired to assist a busy clinician by automating the delivery of blood products based on point-of-care hemostasis testing, continuous coagulation monitoring, and personalized, timely therapeutic delivery according to programmed

knowledge and artificial intelligence[3], the combination of which will realize a cybermedical system[4]. Steps toward a practical coagulation cybermedical system, Fig. 1, have been made for trauma[5], including its process control[6–8] and targeted treatment[9]. However, to truly realize a closed-loop, feedback-based, cybermedical blood-product delivery system, repeated and sequential measurements are needed to elucidate a patient's coagulation patterns, and to capture and predict the time course of clotting dynamics. Conventional coagulation tests like prothrombin time and partial thromboplastin time do not yield sufficient information to guide personalization[9]. Although viscoelastic tests provide patient-specific clotting information, these assays are

[1]Department of Mechanical and Aerospace Engineering, University of Florida, 527 Gale Lemerand Drive, Gainesville, FL 32611-6250, USA. [2]Department of Surgery, University of Colorado Denver, 12631 East 17th Avenue, Mailstop C305, Aurora, CO 80045-2527, USA. [3]Center for Combat Medicine and Battlefield (COMBAT) Research, Department of Emergency Medicine, University of Colorado Denver, 12401 East 17th Avenue, Mailstop B215, Aurora, CO 80045-2589, USA. [4]J. Crayton Pruitt Family Department of Biomedical Engineering, University of Florida, 1275 Center Drive, Gainesville, FL 32611-6131, USA. [5]Department of Agricultural and Biological Engineering, University of Florida, 1741 Museum Road, Gainesville, FL 32611-0570, USA. [6]Genetics Institute, University of Florida, 2033 Mowry Road, Gainesville, FL 32610-3610, USA. [7]Present address: ASML, 17075 Thornmint Court, San Diego, CA 92127-2413, USA. e-mail: amormenezes@ufl.edu

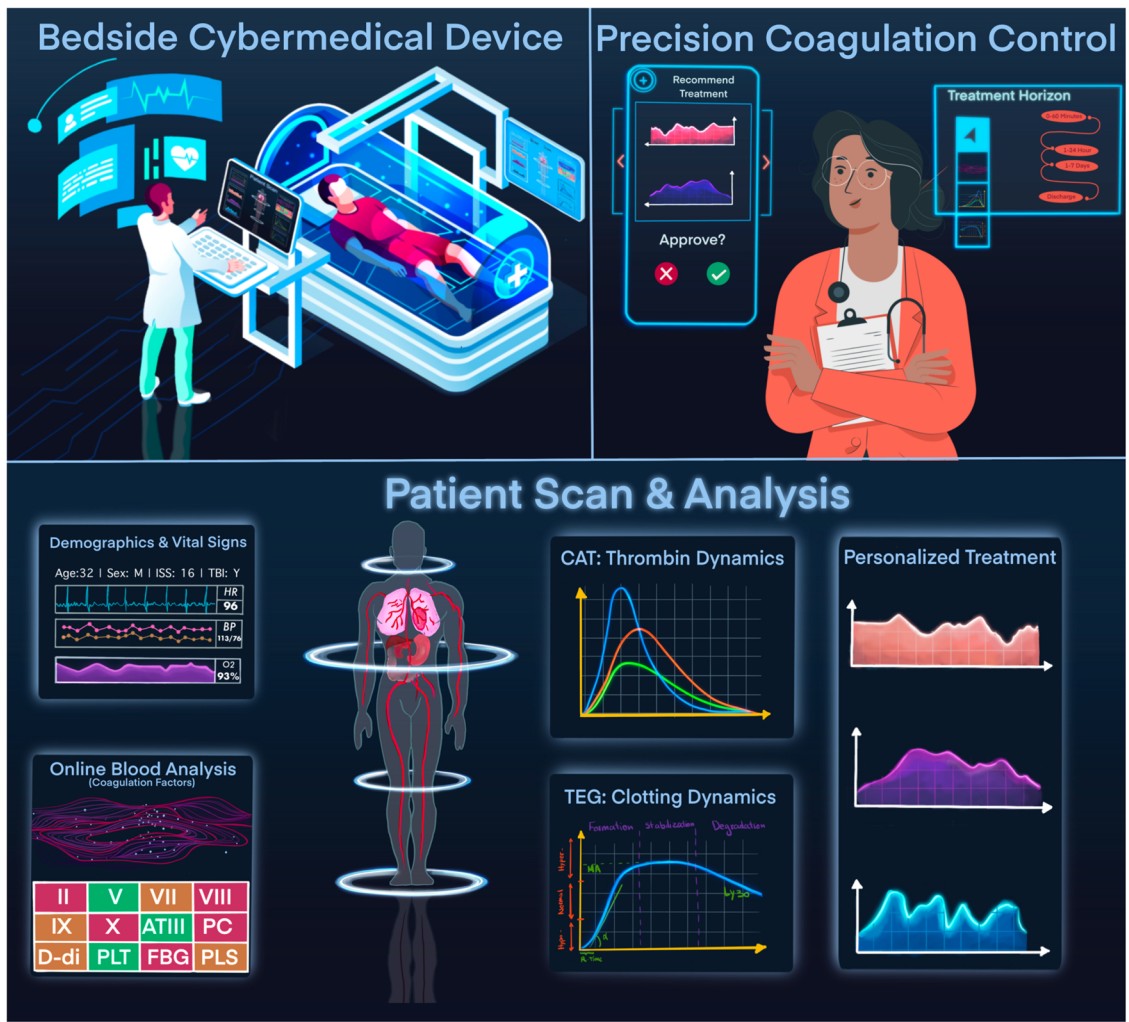

**Fig. 1 | A coagulation cybermedical system.** Elements include: (1) Top left: point-of-care hemostasis testing, e.g., by measuring the concentration of blood sample components like factors II, V, VII, VIII, IX, X, antithrombin (ATIII), protein C (PC), fibrinogen (FBG), plasmin (PLS), fibrin degradation product D-dimer (D-di), and platelets (PLT). This illustration is a lightly modified version of purchased VectorStock image 25805645. (2) Top right: continuous coagulation monitoring, e.g., by measuring thrombin generation via the Calibrated Automated Thrombogram (CAT)[50] and clot strength via Thromboelastography (TEG)[12]. The doctor semi flat illustration is courtesy of Storyset. (3) Bottom: personalized, timely delivery of blood products according to programmed knowledge and artificial intelligence, e.g., in the form of clinician decision support provided on a hand-held device, or autonomously, using a feedback control system[6,7].

quite time-consuming at about an hour per run, which is in addition to sample preparation time, and even newer rapid versions of these assays take at least 30 min[10]. Such delays preclude treatment protocols from being dynamic, and prevent the automation of interventions due to few and incomplete patient-specific measurements. Delays can also reduce treatment efficacy, especially in hemorrhage[11].

Thromboelastography (TEG) and Rotational Thromboelastometry (ROTEM) are two pervasive and similar viscoelastic coagulation assays that provide information on clot development, stabilization, and dissolution reflecting in vivo hemostasis[12]. Clot mechanical properties like viscoelasticity are critical to clot function since overclotting blocks vessels and leads to thrombosis, while weak clots do not close an injury enough to stop bleeding[13,14]. These viscoelastic measures indicate the clot strength of a blood sample with the aid of an immersed pin that measures changes in rotational force as the sample blood coagulates. Clot strength measurements can identify patients at risk for thromboembolic events[15], and can improve patient outcomes in numerous conditions: cardiac[16], vascular[17], and hip fracture surgeries[18,19]; liver cirrhosis and transplants[20–22]; coronavirus disease 2019 (COVID-19)[23]; intensive care unit stays[24]; sepsis[25]; obstetrics[26–28]; diabetes[29]; hemophilia[30]; and trauma[31], where viscoelastic measurements can guide goal-driven patient treatments[32–34]. In all these conditions, blood viscoelastic measurements provide clinical insight into the delivery of blood products such as transfusions (e.g., fresh frozen plasma[35]), blood protein concentrates (e.g., factor VII[30]), pharmacological agents (e.g., tranexamic acid[36]), and anticoagulant treatments (e.g., heparin[23]).

Models that substitute for the TEG/ROTEM assay and make viscoelastic clot strength predictions based on coagulation factor concentrations are a viable alternative. This is because the concentrations of multiple coagulation factors in a blood sample can be measured rapidly (within about five minutes), and because coagulation factor deficiencies affect plasma coagulation kinetics as captured by TEG[37]. Thereafter, control algorithms can leverage these models to generate clotting predictions from patient blood sample measurements, and provide frequent, personalized, and dynamic treatment recommendations to move a patient along a desired recovery trajectory. Although several viscoelastic models of blood flow[38] and blood coagulation[39,40] exist, these models are highly mechanistic, computationally heavy, and unsuited to a cybermedical input-output type of control system application. A recent ROTEM modeling advance[41] may enable the estimation of an important coagulation factor, factor XIII, in a sample, and may also enable the estimation of other sample

coagulation factor concentrations. However, the forward problem of predicting a ROTEM curve from sample coagulation factor concentrations remains open. A promising TEG model[42] cannot yet identify coagulation factor concentration deficits.

Hence, there is a need for a model that replaces the TEG / ROTEM assay and that quickly provides the viscoelastic parameters that clinicians use to indicate patient coagulation status and guide treatment. These parameters are derived from a TEG or ROTEM output curve and consist of: reaction R time, which is the time from the start of the reaction until a clot strength of 2 mm amplitude is measured on the y-axis; K time, which is the time immediately after the R time until a clot firmness of 20 mm amplitude is achieved; the angle $\alpha$, a proxy for the slope of the curve, which reflects the initial rate of clot formation; the maximum amplitude MA, which indicates the maximum clot strength; and Ly30, which is a measurement of clot lysis, or degradation, 30 min after the MA time.

Here, from a control-oriented, phenomenological perspective, we develop quick viscoelastic, dynamic coagulation models that accurately predict the above parameters, toward future deployment for control-oriented interventions of coagulation factor concentrations. We show that model-based prediction of coagulation state for fast (<1 s), frequent, automated, and tailored treatment is feasible. Such predictions can potentially obviate the need for TEG assays in numerous disease conditions, reduce the time to ascertain clotting in patients, and thus increase the targeting of interventions at the point-of-care. Our intellectual contributions include: (i) a novel, simple model that captures the viscoelastic contributions of quickly- and easily-measurable coagulation factor concentrations in plasma; (ii) a second new model to express viscoelastic clot formation, stabilization, and degradation in whole blood; (iii) an identification of platelet and platelet inhibition effects on clot strength; (iv) mappings from our viscoelastic blood models to typical clot measurement parameters for different TEG assays (Functional Fibrinogen, Citrated Native, Platelet Mapping, and Rapid TEG); and (v) a comparison to the generalized Maxwell viscoelastic model to mechanistically justify our new identified models.

## Results

### Modeling approach

We first articulated our core modeling idea in a conference paper[43] and tested preliminary idea implementation on a small trauma patient dataset of whole blood TEG assays. However, the ensuing viscoelastic curve prediction errors were large. This archival article differentiates itself by including: substantial model revisions to improve prediction errors; experimental model validation; model verification on a larger and different trauma patient dataset; a determination of model-to-parameter mappings for additional types of TEG assays; the incorporation of known hemostatic understanding; and the placement of our work in a broader context beyond trauma coagulopathies.

Taken together, our results capture both stages of the blood coagulation mechanism to stop bleeding, i.e., hemostasis, which are reflected in viscoelastic clot strength measurements[44]: (i) primary hemostasis: the formation of a weak platelet plug; and (ii) secondary hemostasis: the formation of a clot through a fibrin network that stabilizes this platelet plug. Primary hemostasis is the initial response upon injury, when damaged endothelium (the tissue that forms a single layer of cells) exposes von Willebrand factor (vWF) and triggers inflammatory mediators. Subsequently, blood platelets adhere to the areas with exposed vWF. Thrombin (factor IIa, where the supplementary "a" denotes an activated coagulation factor) facilitates platelet attachment to the vWF and any circulating fibrinogen to form a weak platelet plug.

Thrombin is the end product of the coagulation cascade[45], a network of coagulation factor biochemical reactions. Secondary hemostasis involves these coagulation factors stabilizing the weak platelet plug. Once the coagulation cascade is triggered by the release of tissue factor (TF) upon injury, thrombin is generated to enhance

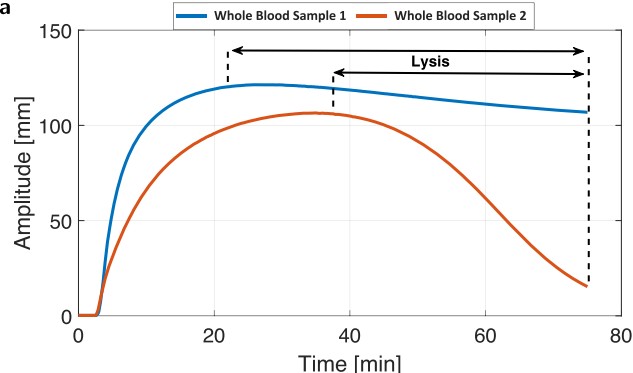

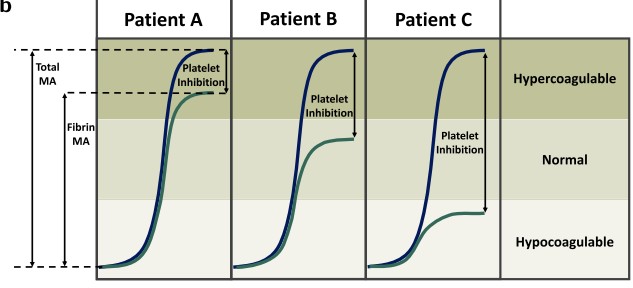

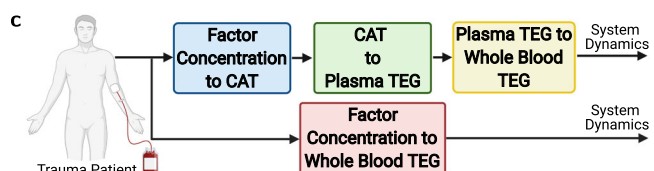

**Fig. 2 | Key elements of viscoelastic clot strength curves and approaches to modeling them.** Source data are provided as a Source Data file. **a** Experimental TEG curves from two whole blood trauma patient samples show differences in lysis. **b** Clot strength (total MA, navy) has secondary hemostasis (fibrin MA, olive) as an underlying component, and their difference can be interrogated with platelet inhibition, the amount of which indicates the amount of primary hemostasis. A hypercoagulable patient can have any one of hypercoagulable, normal, or hypocoagulable secondary hemostasis. **c** This article presents two parallel approaches to predict TEG output from measured coagulation factor concentrations in a patient blood sample. The top path first predicts thrombin generation (blue block), which is then used to predict viscoelasticity (green block), which is thereafter corrected for platelet effects (yellow block). The bottom path predicts whole blood viscoelasticity directly from coagulation factor concentrations (red block). Created with BioRender.com.

platelet action as described above, activate factor XIII into factor XIIIa, and convert fibrinogen into fibrin. This fibrin then forms the cross-linked mesh that binds and stabilizes the weak platelet plug to stop hemorrhage. As a procoagulant, thrombin also catalyzes other coagulation-related reactions, like the activation of factors V, VIII, XI, and protein C (PC), which in turn regulate thrombin generation[46]. As an anticoagulant, thrombin binds to thrombomodulin to initiate fibrinolysis, i.e., clot degradation[47].

This fibrinolysis is reflected in viscoelastic clot strength measurement curves, Fig. 2a, via Ly30. Fibrinolysis dissolves a blood clot to prevent it from becoming large and problematic[48]. Plasminogen, a key fibrinolysis component, is weaved into a blood clot during its formation. Tissue plasminogen activator (tPA, an enzyme that is slowly released by the damaged endothelium of blood vessels) and factors XIa and XIIa convert plasminogen to plasmin. This plasmin breaks down fibrin strands, resulting in blood clot dissolution and fibrin degradation products like D-dimer[49].

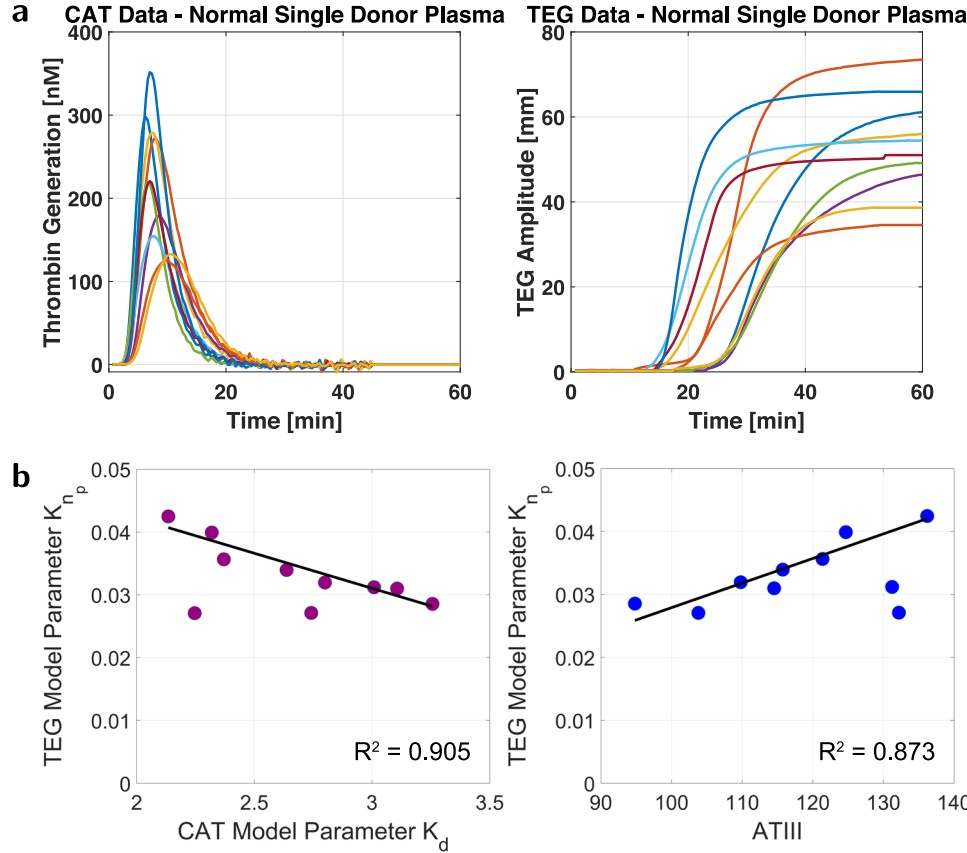

**Fig. 3 | CAT and TEG plasma experiments and model relationships.** Source data are provided as a Source Data file. **a** Experimental CAT curves of ten normal plasma samples, Dataset (7) Supplementary Fig. 1, and their experimental TEG curves, which indicate clot strength from protein concentrations alone. **b** Linear correlations exist between TEG model parameters, CAT model parameters, and coagulation factor concentrations. A linear relationship between $K_{n_p}$ in (1) and $K_d$ in the model from ref. 5 has $R^2 = 0.905$. A linear relationship between $K_{n_p}$ in (1) and ATIII concentration has $R^2 = 0.873$.

Both primary and secondary hemostasis effects are merged in the MA parameter, Fig. 2b, although the constituent fibrin MA value that indicates secondary hemostasis is not immediately apparent from this total MA parameter. Because platelet contributions to clot strength complement coagulation factor contributions to clot strength, patients with the same total MA can have different constituent clot strength contributions from platelets (primary hemostasis) and fibrin MA (secondary hemostasis). The amount that platelets contribute to clot strength can be isolated by platelet inhibition, which can reveal secondary hemostasis effects that can be quite different from the whole blood viscoelastic indication (e.g., Patient C in Fig. 2b has a hypercoagulable whole blood viscoelastic curve but a hypocoagulable fibrin clot strength curve). Providing a treatment that increases clot strength by using a fixed amount of added platelets can therefore yield very different coagulation outcomes depending on the fibrin MA that a patient starts with. Hence, it is very important to separately quantify platelet and coagulation factor contributions to viscoelastic clot strength in a prediction model.

Figure 2c illustrates two parallel paths to obtaining the dynamics of viscoelastic clotting in a blood sample. The top path systematically deduces secondary hemostasis (blue and green blocks) and then primary hemostasis (yellow block), by first using piece-by-piece models for thrombin generation from other coagulation factors (blue block), then predicting their contribution to overall viscoelasticity (green block), and then accounting for platelet effects in whole blood viscoelasticity (yellow block). The bottom path constructs a viscoelastic model of clotting in whole blood directly from coagulation factor concentrations (red block).

In the top path, the thrombin generation piece can come from a model that replaces the Calibrated Automated Thrombogram (CAT)[50], which is another time-consuming coagulation assay that is applied to the plasma component of a whole blood sample to measure the concentration time-history of thrombin upon TF addition, as in the coagulation cascade. An example model from our prior work[5] showed the importance of PC on inferred coagulation dynamics, as well as the impact that antithrombin III (ATIII) had on a delay-associated model parameter, $K_d$. This inference captures the known mechanistic biology, because ATIII is an anticoagulant that slows down thrombin production during coagulation. We next develop models for the green, red, and yellow blocks in Fig. 2c.

The datasets used in this article and their contributions to model development are organized in Supplementary Fig. 1. The characteristics of the trauma patients in these datasets are summarized in Supplementary Fig. 2.

**Platelet-poor plasma viscoelasticity and its relationship to TEG functional fibrinogen**

We performed TEG experiments on platelet-poor plasma (PPP) samples of blood from ten normal individual donors (Dataset (7), Supplementary Fig. 1) to isolate the effects of coagulation factors following thrombin generation (the second green block in the top path of Fig. 2c). The results of CAT and TEG assays for each plasma sample are in Fig. 3a.

Application of the Akaike Information Criterion[51] to this experimental data suggested that a simple, second-order viscoelastic model from an input of thrombin (CAT data) to an output of clot strength (TEG curve) is sufficient to capture observed behavior without overfitting. We propose a second-order plasma TEG model that consists of

three elements: a first-order system to capture the step-like increase in clotting activity, a first-order integrator to hold this value over time, and a time delay for a horizontal shift. Below, we state this model in the controls-oriented frequency domain $s$ instead of the time domain $t$; the implication is that we assume that the underlying dynamical system is linear and time-invariant[52]. This is a reasonable assumption despite nonlinear coagulation dynamics[53] and possibly time-varying parameters, because our prior work[9] showed that a linear, time-invariant model for coagulation is experimentally justified at the typical input value of 5 pM TF. The CAT curves for Fig. 3a are for a 5 pM TF input.

Our input-output transfer function model (from CAT to Plasma TEG) is:

$$G(s) = \frac{P(s)}{Y(s)} = \frac{K_{n_p}}{s(K_p s + 1)} e^{-K_{d_p} s}, \quad (1)$$

where $Y(s)$ is the thrombin generation input, and $P(s)$ is the plasma TEG output. This model has three parameters: $K_{n_p}$ defines the maximum TEG amplitude, equivalently clot firmness; $K_p$ is a time constant of clot formation kinetics; and $K_{d_p}$ is a delay term for clot initiation. Changes in these parameters change viscoelastic properties between samples.

Model (1) accurately captures plasma viscoelasticity. Using the MATLAB Simulink Design Optimization (SDO) toolbox, unique parameters of model (1) were fit for each sample in Fig. 3a, with an input of thrombin (CAT) data and an output of clot strength (TEG) data. The fit parameters minimized the least square error between model simulation and experimental data using the trust region reflective algorithm and a solver tolerance of $10^{-9}$. The mean $R^2$ value of the fits was $R^2 = 0.985$, which confirms model capacity to capture the observed data.

We identified correlations between the fit parameters for TEG data and those of our published model[5] for CAT data. Figure 3b shows that coefficient $K_{n_p}$ in model (1) is linearly correlated with coefficient $K_d$ in our CAT model[5] (the plotted line of best fit was obtained by excluding two outliers; this line has $R^2 = 0.905$). Since ATIII is a primary driver of $K_d$[5], we hypothesized a correlation between parameter $K_{n_p}$ and ATIII. We confirmed the existence of a linear relationship, Fig. 3b (the plotted line of best fit was again obtained by excluding two outliers; this line has $R^2 = 0.873$). Consequently, predicting the parameters of model (1) from coagulation factor concentrations is feasible. We developed estimates of these parameters using the matching pursuit algorithm[54] on concentrations of factors II, V, VII, VIII, IX, X, and ATIII. The percent error results and the high quality of these estimates are in Fig. 4a, b, respectively.

Since this test performance was on the same dataset as that used for learning, and the number of samples in our dataset was limited, we applied two validation techniques. First, we performed fivefold cross-validation to bootstrap our data while checking predictive performance. That is, we estimated parameters using the matching pursuit algorithm on 80% of the data, and validated performance on the remaining 20% of dataset samples, repeating the process five times. The mean percent errors are in Fig. 4c. These mean percent errors were calculated by comparing the estimated model properties to the actual values obtained from experimental fits. This table quantifies good prediction of plasma TEG curves, with the entire dataset clearly required for delay time prediction.

We additionally validated model (1) using Dataset (8), Supplementary Fig. 1, consisting of plasma TEGs of five normal donors who were different from those constituting the dataset that was used in model parameter estimation training, Dataset (7). Plasma TEG graph estimates for Dataset (8) using each plasma sample's coagulation factor concentrations are in Fig. 5a, and the associated mean relative error of the estimated graph properties in Fig. 5b confirms the accuracy of the model. We anticipate improving our MA time predictions with more training data.

But our MA value predictions for plasma TEGs, the fibrin MA values, are quite accurate. A fibrin MA value is equivalent to the experimental MA value that can be obtained from the TEG Functional Fibrinogen (FF) assay, which is a whole blood TEG assay that evaluates fibrinogen contributions to clot strength by blocking platelet contributions using a potent inhibitor. Because the clot strength of platelet-poor plasma is proportional to functional fibrinogen concentration[37], our predicted MA value can be used as an indicator of secondary hemostasis.

To show this, consider the following transformation of TEG FF MA to a shear elasticity value, $G_f$ [dynes/centimeter$^2$, i.e., 0.1 Pascal][55]:

$$G_f = 5000 \times \frac{MA}{100 - MA}. \quad (2)$$

Figure 5c demonstrates a linear map from $G_f$ to the TEG machine-reported functional fibrinogen level (FLEV), $R^2 = 0.990$, for $G_f$ transformed from the TEG FF MA available for 63 of the 97 trauma patients (Dataset (6), Supplementary Fig. 1). A similar linear map from $G_f$ to laboratory-measured fibrinogen concentration exists for the same dataset, Fig. 5d. Therefore, even if coagulation is abnormal, it is possible to replace the TEG FLEV value and the TEG FF assay parameters by leveraging quickly-measurable coagulation factor concentrations, predicting MA using model (1), and then using (2) to predict TEG FLEV and functional fibrinogen, and hence secondary hemostasis.

## Whole blood viscoelasticity and its relationship to TEG citrated native

Plasma TEGs cannot show two key clot strength elements. First, because platelets are removed from a plasma sample, platelet contributions to whole blood viscoelasticity and overall clot strength (for example, the TEG curves in Fig. 6a) are missing. Thus, the TEG curve of a plasma sample, Fig. 5a, has a lower MA compared to the TEG curve of its whole blood counterpart, Fig. 7a. Second, drivers of fibrinolysis are also not present in a plasma sample. Accordingly, the typical decays of whole blood TEG curves, Fig. 2a, are omitted, and the plasma TEGs in Figs. 3a and 4b are constant after reaching MA, since the formed clot remains intact over time.

We extend plasma TEG model (1) to capture the effects of both platelets and fibrinolysis (the red block in the bottom path of Fig. 2c). We assume that platelet contributions can be captured by a scaling gain to increase MA, an assumption that we confirm later in this article. We propose that fibrinolysis requires a model of identical structure to clot initiation (i.e., a second-order system) but in the opposite direction for decay. Hence, a two-term viscoelastic frequency domain model to capture full clot dynamics is:

$$G(s) = \frac{W(s)}{U(s)} = \frac{K_{n_1}}{s(K_{p_1} s + 1)} e^{-K_{d_1} s} - \frac{K_{n_2}}{s(K_{p_2} s + 1)} e^{-K_{d_2} s}, \quad (3)$$

where $K_{n_1}$, $K_{n_2}$, $K_{p_1}$, $K_{p_2}$, $K_{d_1}$, and $K_{d_2}$ are positive parameters, $U(s)$ is a 5 pM impulse TF input as before, and $W(s)$ is the TEG whole blood model output. The first term of model (3) has the structure of model (1) to represent the effects of coagulation factors, with $K_{n_1}$ including the scaling effects of platelets. In this term, $K_{d_1}$ is the time to initial clot formation, and $K_{p_1}$ relates to the speed of clot formation. The second term of model (3) represents fibrinolysis, reversing clot formation effects from the first term due to the negative sign in front of $K_{n_2}$. In this term, $K_{d_2}$ is the time to fibrinolysis start, and $K_{p_2}$ relates to the speed of clot breakdown.

With MATLAB SDO, we fit model (3) to 15 patients of an experimental TEG dataset of 24 trauma patient whole blood samples, Dataset (10), Supplementary Fig. 1, to obtain unique parameters for each sample (a mean $R^2 = 0.9995$ attests to model suitability). These 15

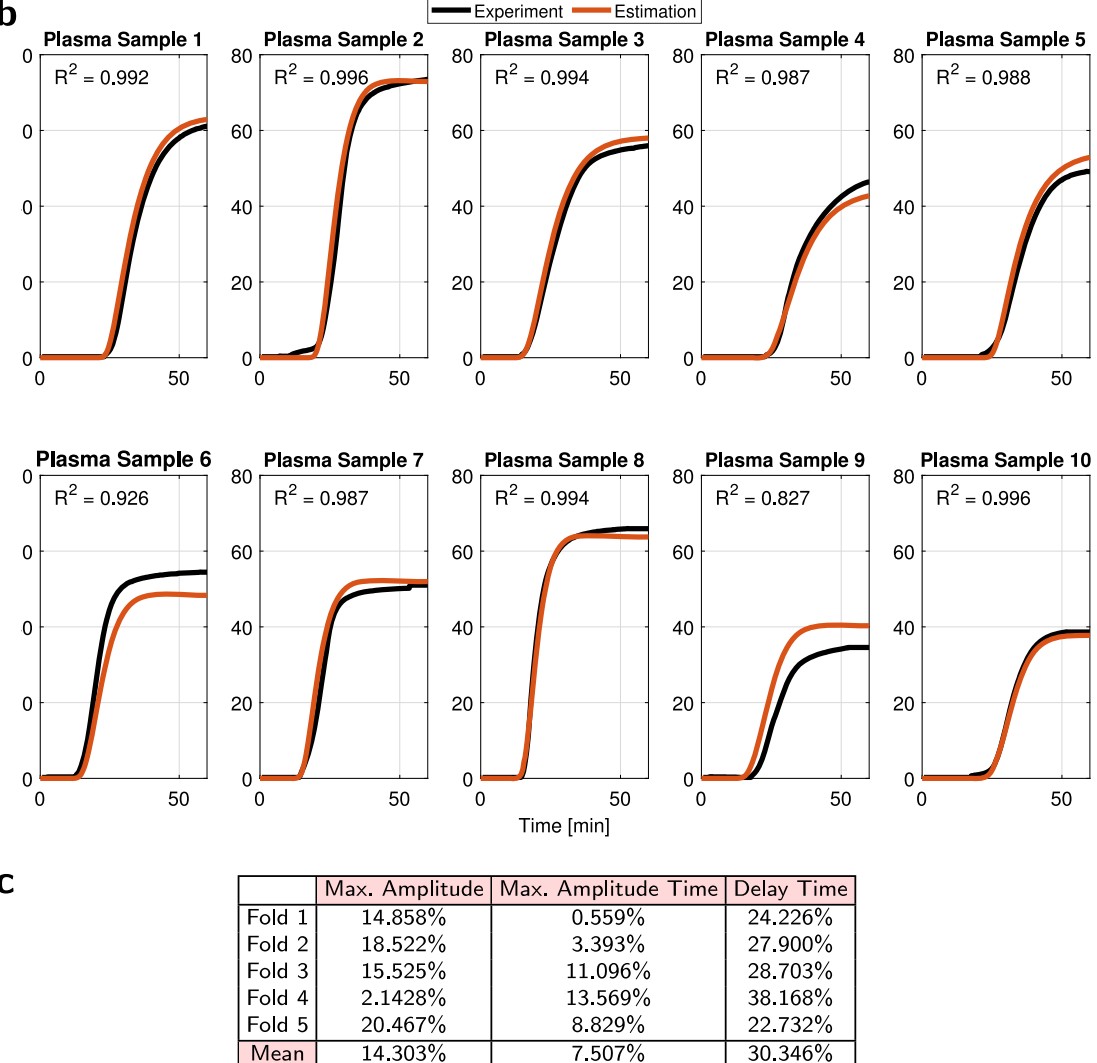

**a**

|  | Max. Amplitude | Max. Amplitude Time | Delay Time |
|---|---|---|---|
| Mean Error | 5.774% | 11.443% | 9.694% |

**b**

**c**

|  | Max. Amplitude | Max. Amplitude Time | Delay Time |
|---|---|---|---|
| Fold 1 | 14.858% | 0.559% | 24.226% |
| Fold 2 | 18.522% | 3.393% | 27.900% |
| Fold 3 | 15.525% | 11.096% | 28.703% |
| Fold 4 | 2.1428% | 13.569% | 38.168% |
| Fold 5 | 20.467% | 8.829% | 22.732% |
| Mean | 14.303% | 7.507% | 30.346% |

**Fig. 4 | Dataset (7) plasma TEG curves and model estimates.** Source data are provided as a Source Data file. **a** TEG model (1) relative error. **b** Experimental TEG curves from plasma samples compared to TEG model predictions confirm the efficacy of model (1), with a mean prediction accuracy of $R^2 = 0.969$. **c** Fivefold cross-validation of plasma TEG model (1).

patients had complete coagulation factor concentration data. We developed parameter estimates using the matching pursuit algorithm[54] on measured concentrations of coagulation factors. We used the concentrations of factors II, V, VII, VIII, IX, X, ATIII, and PC to estimate $K_{d_1}$ and $K_{p_1}$ due to the role of these coagulation factors in secondary hemostasis. We used the concentrations of these eight coagulation factors as well as the concentrations of fibrinogen and platelets to estimate $K_{n_1}$, so that the parameter accounts for both primary and secondary hemostasis. We additionally used D-dimer (fibrin degradation product) measurements when estimating parameters of the second term of model (3). Figure 6a shows satisfactory estimation of clot strength, stability, and degradation, and Fig. 6b quantifies the accuracy of model (3).

We again applied two validation techniques, fivefold cross-validation and a dataset not used for training. We used fivefold cross-validation to bootstrap Dataset (10) by splitting it into five subsets (four for training and one for validation) to estimate clotting properties. The mean relative error for TEG graph properties in each fold and the overall mean error is in Fig. 6c. This table quantifies good prediction of whole blood TEG curves, with MA time prediction improvements possible with more training data. We also validated model (3) with Dataset (8), Supplementary Fig. 1, which had whole blood TEGs in addition to plasma TEGs for the five normal donors, and these donors were different individuals from the trauma patients who constituted the dataset that was used in model parameter estimation training, Dataset (10). Whole blood TEG graph estimates for Dataset (8) are in Fig. 7a, and the associated mean relative error of the estimated graph properties in Fig. 7b confirms the accuracy of the model.

Our model (3) can replace the TEG Citrated Native (CN) assay, which evaluates clot formation, stabilization, and degradation in whole blood samples. This is true even if coagulation is abnormal. As Fig. 7c shows with red lines of best fit for the 24 trauma patients in Dataset (10), fitted $K_{n_1}$ is directly related to MA, $R^2 = 0.838$; fitted $K_{d_1}$ is directly related to R time, $R^2 = 0.989$; fitted $K_{p_1}$ is inversely related to $\alpha$ angle, $R^2 = 0.687$; and the reduction in area under the curve (AUC), as computed from the fibrinolysis term in the model, is directly related to

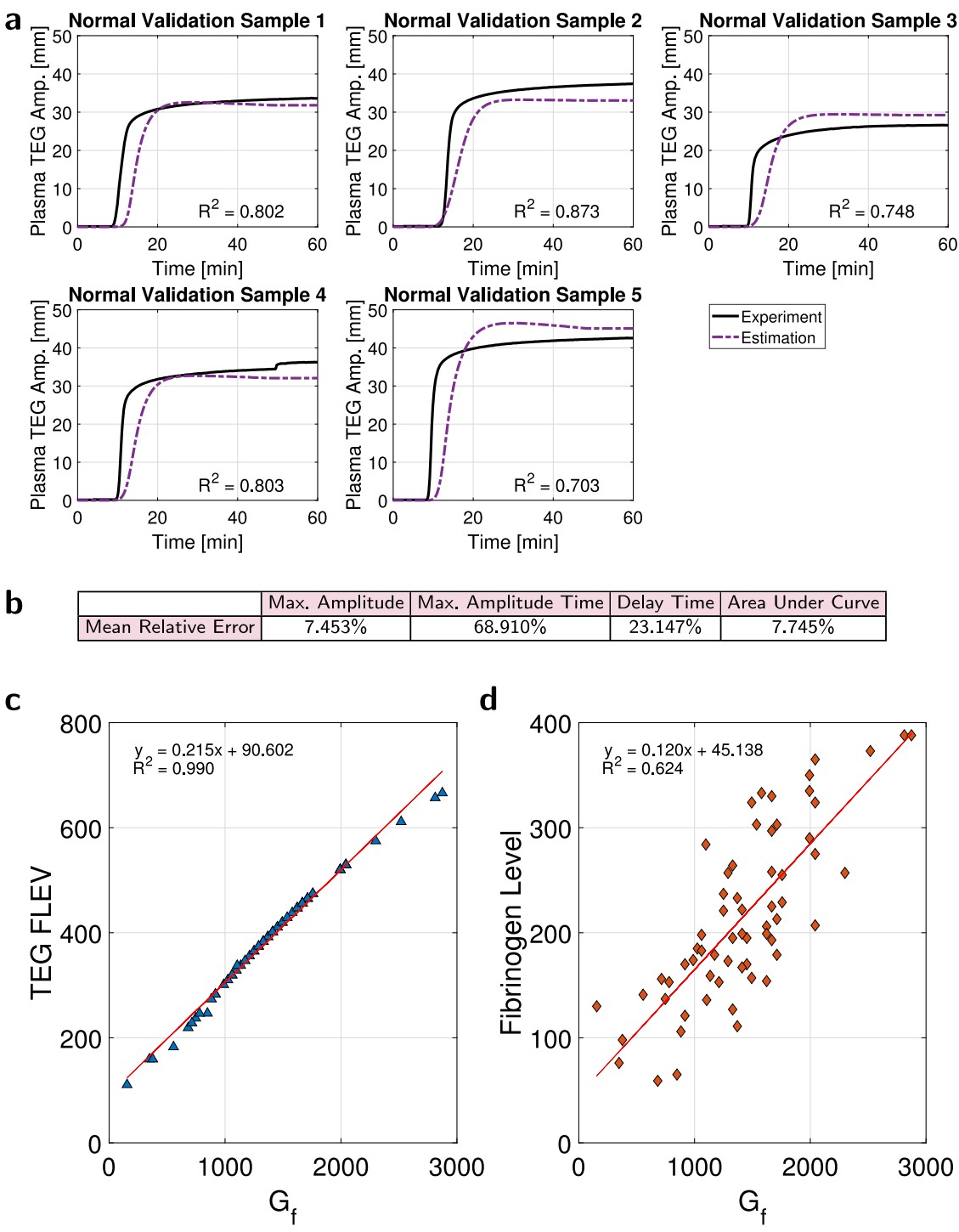

**Fig. 5 | Validation of plasma TEG model (1), and its connection to the TEG FF assay.** Source data are provided as a Source Data file. **a** Predictions of experimental TEG curves of five normal plasma samples using coagulation factor concentrations (Dataset (8), Supplementary Fig. 1, which was not used in model estimation training) confirm the efficacy of model (1), with a mean prediction accuracy of $R^2 = 0.786$. **b** Prediction mean relative error, with low errors on MA. **c** The parameter $G_f$, a transformation of MA, available for 63 of the 97 trauma patient samples (Dataset (6), Supplementary Fig. 1), is nearly perfectly correlated with TEG reported fibrinogen level (FLEV) from the FF assay, $R^2 = 0.990$, and **d** correlated with the measured fibrinogen level in these patients, $R^2 = 0.624$, which is an indicator of secondary hemostasis. It follows that plasma TEG model predictions of MA can replace the TEG FF assay and be a viable indicator of secondary hemostasis even in abnormal clotting conditions.

Ly30, $R^2 = 0.931$. Therefore, it is possible to replace the TEG CN assay parameters by leveraging quickly-measurable coagulation factor concentrations, predicting the parameters of model (3), and then using these parameters to predict the TEG CN parameters (MA, R time, $\alpha$ angle, and Ly30), thereby substantially reducing long experiment times.

## Platelet viscoelasticity and its relationship to TEG Platelet Mapping

We next show that platelet contributions to clot strength scale up the fibrin MA of model (1) (thus constituting the third yellow block in the top path of Fig. 2c). The TEG Platelet Mapping (PLT) assay enables a quantitative analysis of platelet function by relating

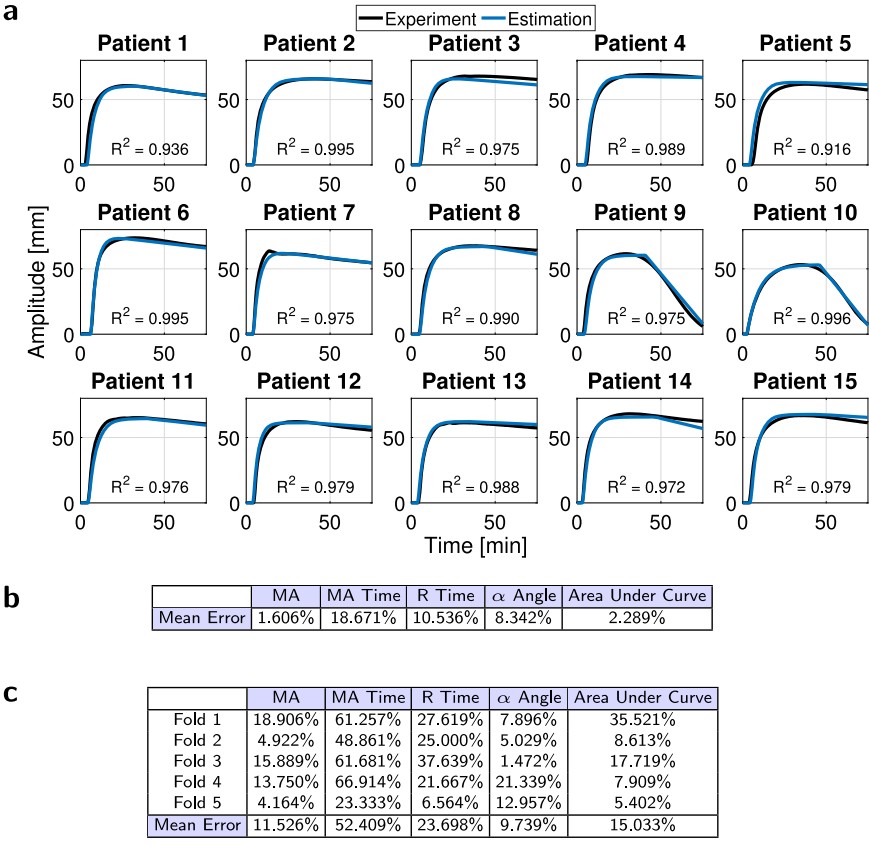

**Fig. 6 | Dataset (10) whole blood TEG curves and model estimates.** Source data are provided as a Source Data file. **a** Experimental TEG curves on whole blood samples from 15 trauma patients compared to TEG model predictions confirm the efficacy of model (3), with a mean prediction accuracy of $R^2 = 0.976$. **b** TEG model (3) relative error. **c** Fivefold cross-validation of whole blood TEG model (3).

percent platelet inhibition to an individual's maximum uninhibited platelet function. We used a dataset of 48 trauma patients, Dataset (11), Supplementary Fig. 1, for whom we had TEG FF, CN, and PLT assays, as well as platelet count measurements. Figure 8a compares the MA from the CN and PLT assays. The red line of best fit, $R^2 = 0.487$, shows that these two measurements are linearly correlated with a nearly one-to-one map (e.g., a whole blood sample will have an MA of 60 mm on both the CN and PLT assays). Figure 8b compares the MA from the FF and PLT assays. The red line of best fit through the origin, $R^2 = 0.583$, depicts that a linear relationship between these two measurements exists. Hence, an alternate way to estimate CN MA via $K_{n_1}$ in model (3) is to predict the fibrin MA from quickly-measurable coagulation factor concentrations using model (1), and then multiply this fibrin MA by the slope of the fitted line from Fig. 8b.

Because Dataset (11) did not have coagulation factor concentration data, and Datasets (8) and (10) did not have TEG PLT assay data, we validated our above scaling claim using platelet count measurements. First, adenosine diphosphate (ADP) is a platelet agonist that inhibits platelet activity, thereby reducing clot strength. The TEG PLT assay uses ADP to indicate percent platelet inhibition. Fig. 8c illustrates the inhibitory effects of ADP in reducing clot strength, where % MA Reduction $= \frac{\text{MA}_{\text{CN}} - \text{MA}_{\text{FF}}}{\text{MA}_{\text{CN}}}$. To evaluate platelet function, we next identified a relationship between uninhibited (activated) platelets (which are the total number of measured platelets multiplied by $1 - \%$ ADP inhibition) and the effect that these uninhibited platelets had on the amplification from plasma MA to whole blood MA (i.e., $\frac{\text{MA}_{\text{CN}}}{\text{MA}_{\text{FF}}}$),

Fig. 8d. The line of best fit through the origin has $R^2 = 0.467$. Figure 8d shows that: (i) MA amplification is directly related to uninhibited platelets; (ii) the scale factor is approximately constant for most measurements (platelet counts between 50 and 150), which validates our above scaling claim; and (iii) the scale factor is not constant but varies linearly when evaluated over a large range of platelet counts (from 0 to more than 300). Figure 8c, d also confirms that we can estimate the effect of ADP inhibition in different patients by using a combination of: their platelet counts, fibrin MA from model (1) calculated using their coagulation factor concentrations, and whole blood MA from model (3).

**Predicting rapid TEG parameters from TEG CN parameters**

Since we have previously shown how to predict the parameters of the TEG CN assay from quickly-measurable coagulation factor concentrations and model (3), we can extend our prediction capabilities to the Rapid TEG assay variant. Compared to TEG CN, Rapid TEG uses a higher TF concentration to promote faster coagulation cascade initiation and coagulation factor activation. This implies that thrombin generation is increased and consequently that fibrin production and platelet activation is higher. Accordingly, we anticipate that CN MA will slightly increase, Fig. 9a for a dataset of 97 trauma patients (Dataset (6), Supplementary Fig. 1), which is confirmed by the red line of best fit, $R^2 = 0.516$. We also expect that faster clot formation will reduce R time; however, the R time panel in Fig. 9b shows that this value is constant for all Rapid TEG samples (its ratio with CN R time changes linearly with CN R time) at $1.107 \pm 0.089$ (mean ± standard deviation). TF does not have a direct impact on fibrinolysis, and so the

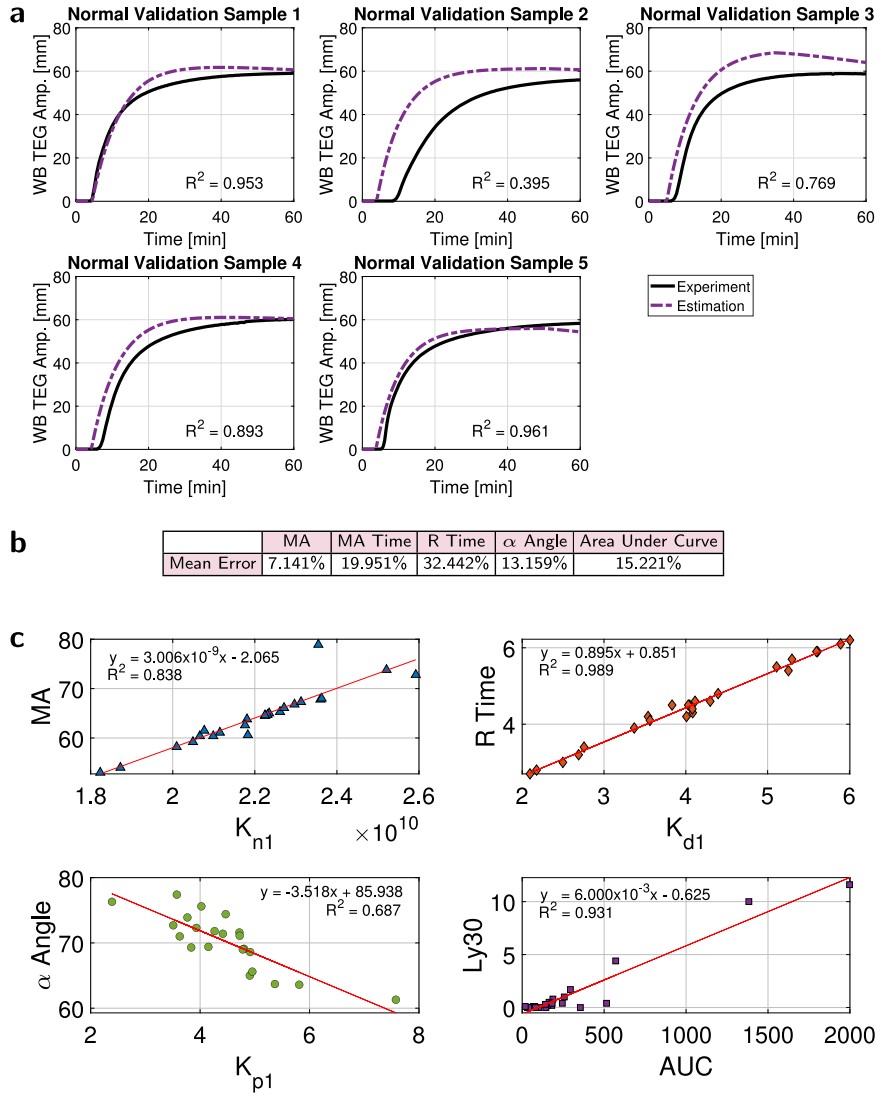

**Fig. 7 | Validation of whole blood TEG model (3), and its connection to the TEG CN assay.** Source data are provided as a Source Data file. **a** Predictions of experimental TEG curves of five normal whole blood samples using coagulation factor concentrations (Dataset (8), which was not used in model estimation training) confirm the efficacy of model (3), with a mean prediction accuracy of $R^2 = 0.794$. **b** Prediction mean relative error. **c** Whole blood TEG model predictions can replace

the TEG CN assay even in abnormal clotting conditions. For the trauma patient samples of Dataset (10): fitted $K_{n_1}$ is directly related to MA, $R^2 = 0.838$; fitted $K_{d_1}$ is directly related to R time, $R^2 = 0.989$; fitted $K_{p_1}$ is inversely related to $\alpha$ angle, $R^2 = 0.687$; and the reduction in AUC (computed from the second, fibrinolysis term in the model) is directly related to Ly30, $R^2 = 0.931$.

Ly30 property that indicates clot breakdown remains identical between Rapid TEG, and TEG CN, Fig. 9c, as indicated by the one-to-one red line of best fit, $R^2 = 0.974$. Finally, faster coagulation scales up the Rapid TEG $\alpha$ angle from TEG CN, Fig. 9d, with red line of best fit $R^2 = 0.457$. The point remains that coagulation factor concentrations can help quickly predict Rapid TEG output, by determining TEG CN parameters via model (3) and then modifying these parameters using the relationships identified in Fig. 9.

## Model relationship to the Maxwell–Wiechert model of viscoelasticity

The generalized Maxwell model (or Maxwell-Wiechert model) consists of several parallel Maxwell elements of viscoelastic material, Fig. 10a, where each Maxwell element is a spring and a damper connected in series. This model accounts for slow force relaxation over time. The generalized Maxwell model for linear viscosity[56] is

$$\sigma(t) = \sigma_0\left(1 - e^{-\frac{1}{\tau}t}\right), \tag{4}$$

which is the solution to the stress-strain ($\sigma - \epsilon$) relationship of a Maxwell material $E\dot{\epsilon}(t) = \dot{\sigma}(t) + \frac{1}{\tau}\sigma(t)$, where $\tau$ is the relaxation time constant defined as $\tau \triangleq \frac{\eta}{E}$, $E = \frac{\sigma_0}{\epsilon_0}$ is Young's elastic modulus, and $\eta$ is viscosity.

The delay-free time-domain representation of model (1) is

$$g(t) = K_{n_p}\left(1 - e^{-\frac{1}{K_p}t}\right), \tag{5}$$

which is identical to the generalized Maxwell model (4) above. Hence, models (1) and (3) have a mechanistic viscoelastic interpretation. In the case of coagulation, after thrombin production is stopped, equivalently after a clotting-promotion force is removed, the resultant clot will break down over time at a patient-specific rate.

A complex dynamic modulus $G$ can be used to represent the relations between the oscillating stress and strain. Similarly, a complex dynamic modulus, $E'$, of a material with elastic modulus $E$ (a Maxwell

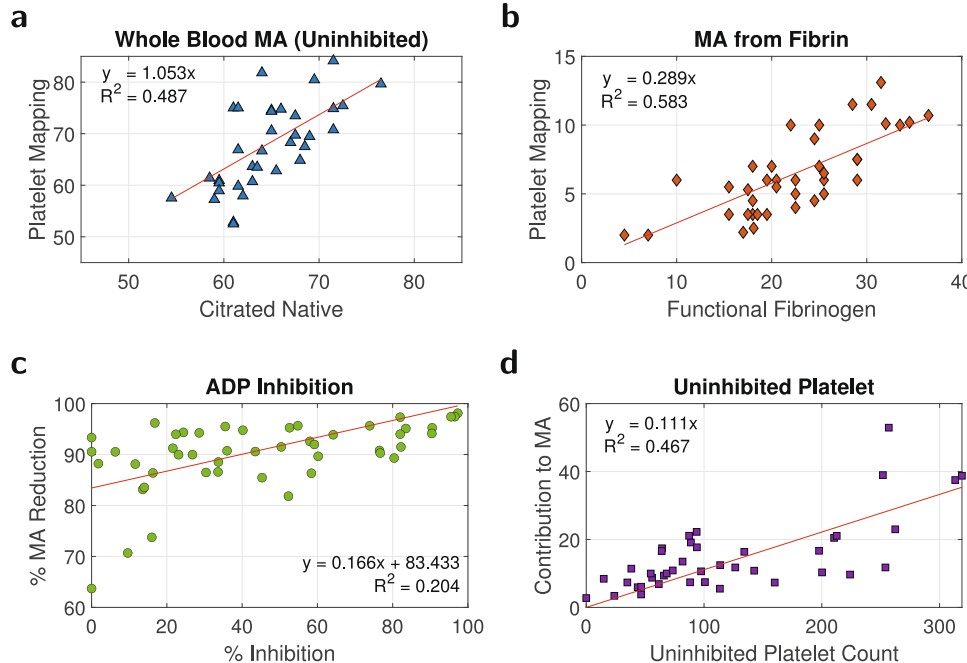

**Fig. 8 | Dataset (11) TEG PLT assay model estimates and validation.** Source data are provided as a Source Data file. **a** Comparison of MA obtained from TEG CN (whole blood contribution to clot strength) and PLT assays, $R^2 = 0.487$. **b** Comparison of MA obtained from TEG FF (fibrin contribution to clot strength) and PLT assays, $R^2 = 0.583$. **c** ADP inhibits platelet activation and consequently reduces clot strength, $R^2 = 0.204$. Here, % MA Reduction = $\frac{MA_{CN} - MA_{FF}}{MA_{CN}}$. **d** The platelet role in clot strength amplification ($\frac{MA_{CN}}{MA_{FF}}$) is directly related to the number of platelets uninhibited by ADP (platelet count × 1 − %ADP inhibition), $R^2 = 0.467$. This plot validates modeling platelet contributions as a scaling "gain" in clot strength over platelet counts between 50 and 150, although the gain can vary linearly with uninhibited platelet count over counts between 0 and more than 300.

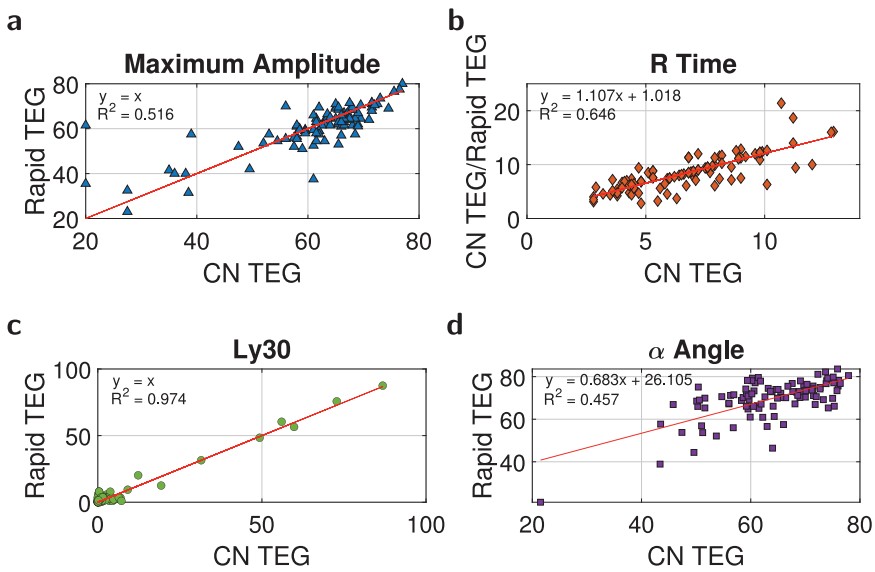

**Fig. 9 | Dataset (6) rapid TEG relationships to the TEG CN assay.** Source data are provided as a Source Data file. **a** Rapid TEG MA is directly related to TEG CN MA, $R^2 = 0.516$. **b** Rapid TEG R time can be estimated as a constant of 1.107, $R^2 = 0.646$. **c** Rapid TEG Ly30 is directly related to TEG CN Ly30, $R^2 = 0.974$. **d** Rapid TEG $\alpha$ angle is directly related to TEG CN $\alpha$ angle, $R^2 = 0.457$.

material) can be expressed with two components[56]:

$$E^* = E_1 + E_2 i, \qquad (6)$$

where $i = \sqrt{-1}$, and storage modulus $E_1$ and loss modulus $E_2$ in (6) are defined as:

$$E_1(\omega) = \frac{\tau^2 \omega^2}{\tau^2 \omega^2 + 1} E; \qquad (7)$$

$$E_2(\omega) = \frac{\tau \omega}{\tau^2 \omega^2 + 1} E, \qquad (8)$$

which has $\omega$ as the oscillation frequency; $\eta$ as the material coefficient of viscosity; and $\tau$ as the relaxation time constant, $\tau \triangleq \frac{\eta}{E}$.

Equations (7) and (8) can be obtained by considering the stress-strain relationship of a Maxwell material $E\dot{\epsilon}(t) = \dot{\sigma}(t) + \frac{1}{\tau}\sigma(t)$, and considering the complex dynamic response where the stress

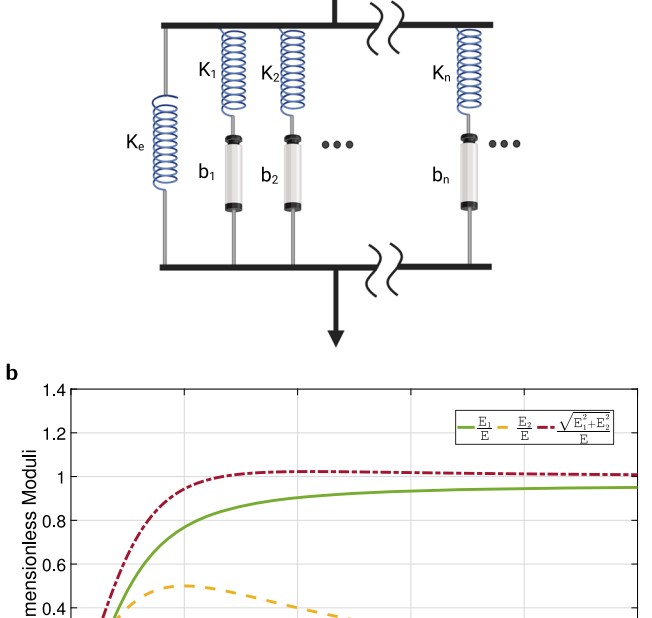

**Fig. 10 | Model interpretation. a** A schematic of the generalized Maxwell model, which is represented by multiple pairs of a purely viscous damper and a purely elastic spring connected in series. The $i$-th viscous damping coefficient is $b_i$, and the $i$-th spring constant is $K_i$. The stress is denoted by $\sigma$. Created with BioRender.com. **b** Maxwell model moduli, with line colors corresponding to the green, yellow, and red blocks in Fig. 2c. The dimensionless storage modulus and the dimensionless loss modulus are the green solid and yellow dashed lines, respectively, and their scaled vector sum is the red dashed-dotted line.

and strain terms have a time dependency of the form $e^{i\omega t}$. Hence, we have:

$$(i\omega)\epsilon_0^* e^{i\omega t} E = \left(i\omega + \frac{1}{\tau}\right)\sigma_0^* e^{i\omega t}. \tag{9}$$

We can write (9) as:

$$
\begin{aligned}
E^* &= \frac{\sigma_0^*}{\epsilon_0^*} = \frac{E(i\omega)}{i\omega + \frac{1}{\tau}} = \frac{E\tau(i\omega)}{1 + i\omega\tau}, \\
&= \frac{(E(i\omega)\tau)(1 - i\omega\tau)}{(1 + i\omega\tau)(1 - i\omega\tau)} = \frac{E\tau\omega i + E\tau^2\omega^2}{1 + \tau^2\omega^2}, \\
&= \frac{\tau^2\omega^2}{1 + \tau^2\omega^2}E + \frac{\tau\omega}{1 + \tau^2\omega^2}Ei.
\end{aligned}
\tag{10}
$$

The final terms of (10) are storage modulus $E_1$, and loss modulus $E_2$ in (7) and (8). The relaxation spectra of dimensionless moduli (7) and (8), which are $\frac{E_1}{E}$ and $\frac{E_2}{E}$, respectively, are in Fig. 10b as a function of dimensionless frequency $\tau\omega$. These moduli indicate whether the elastic ($E_1$) or the viscous ($E_2$) part of the viscoelastic material has the dominant effect at the specific condition.

The above facilitates a mechanistic interpretation of blood coagulation. Thrombin generation can be interpreted as a force and the resultant stress $\sigma$ that drives coagulation. Stopping thrombin generation leads to clot breakdown, a relaxation process. Hence, we can expect that the relaxation time constant $\tau$ in a sample is driven by

coagulation factor concentrations, similar to parameters of models (1) and (3). We hypothesize that loss modulus $E_2$, which is the yellow dashed line in Fig. 10b, represents clot strength effects from platelets. This is because platelets initially contribute to clot strength by forming a weak platelet plug, i.e., primary hemostasis, which can be strengthened by attaching to the fibrin mesh formed from coagulation proteins (crossed-link mesh). However, this platelet plug is unstable without the fibrin crossed-link mesh, and breaks down over time. Storage modulus $E_1$ can be associated with the fibrin mesh contribution to clot strength, i.e., secondary hemostasis. Unsurprisingly, given the comparative similarity of (5) to the generalized Maxwell model, storage modulus $E_1$, the green solid line in Fig. 10b, is comparable to clot strength from coagulation factors in a plasma sample, Fig. 3b.

We compute the magnitude of the complex modulus $E^*$ as $\|E^*\| = \sqrt{E_1^2 + E_2^2}$, the red dashed-dotted line in Fig. 10b. This result is comparable to the whole blood clot strength, which is the product of contribution from all blood components forming the crossed-link fibrin mesh (platelets and fibrin), i.e., both primary and secondary hemostasis. Thus, viscoelastic results in Fig. 10b mechanistically connect to our phenomenological approach to determining the green, yellow, and red blocks in Fig. 2c, and have the same respective colors.

## Discussion

Toward realizing broadly applicable cybermedical coagulation control, which consists of point-of-care hemostasis testing, continuous coagulation monitoring, and personalized, timely therapeutic delivery according to programmed knowledge and artificial intelligence, we propose replacing time-consuming coagulation measurements with model-based computational outputs. We have developed a phenomenological dynamical systems model of plasma sample visocelasticity dynamics using only coagulation factor measurements, to capture the role of plasma components in clot formation. We then expanded this model as a whole blood viscoelastic model to capture the entirety of coagulation: clot formation (including platelet contributions), stabilization, and clot degradation (lysis). Our results showed that all model parameters can be estimated using only quickly-measurable protein concentrations and linear maps, which we confirmed and validated using data from a mix of normal and abnormal clotting systems, the latter from trauma patients. We also detailed how to simply and speedily predict various individual clot strength contributions from coagulation factors, fibrin, and activated platelets, which are currently clinically evaluated using the TEG assays of Functional Fibrinogen, Citrated Native, Platelet Mapping, and Rapid TEG. Importantly, we demonstrated a mechanistic interpretation of our models using the generalized Maxwell model of viscoelasticity.

These results provide direct actionable intelligence for trauma patient care, and constitute a framework leveraging coagulation factor concentrations that can be further developed into a bedside device. Simultaneously, our results also provide important biological insights into the post-injury coagulation milieu. Therefore, our work has the dual benefits of basic biological understanding as well as decision support for the burdened clinician who cares for the severely injured or ill patient.

It takes 5–10 min to measure coagulation factor concentrations using the STA Compact Max®, and the elapsed time for each individual sample estimation averages 0.115 s using a contemporary desktop running MATLAB R2021a on a quadcore Intel Core i7-4790 at 3.40 GHz with 16 GB RAM. Together, the measurement and model run-times in this article provide an outstanding reduction in clotting information delivery time compared to TEG tests of 60 min or more.

We anticipate that model prediction errors will decrease with more sample and TEG data. Because our models show high performance after fivefold cross-validation and also on separate patient datasets, our work can next be verified by clinical studies with a large number of patients and their coagulation factor concentration

measurements, in all settings where viscoelastic clotting measures are insightful. In parallel, our models can be refined by incorporating the effects of adding individual coagulation factor concentrations into samples of normal and trauma patient plasma and whole blood, to confirm model-predicted clotting outcomes and theoretically ground previous literature observations[37]. Such work will also provide insight into treatment feasibility and efficacy from any proposed additional individual or combinatorial protein concentrations.

We envision implementing our advances in an automated, cybermedical treatment device in future. Thus, the work in this article represents a considerable step toward frequent, personalized, and precision coagulation interventions. Given the pervasive importance of thromboinflammation driving critical illness and injury, the results of our work are relevant to other coagulation and inflammation disorders as well, including hemophilia, von Willebrand disease, factor V Leiden, pulmonary embolism, deep vein thrombosis, stroke, sickle cell disease, cancer, and COVID-19.

## Methods

### Coagulation factor concentration measurements

Coagulation factor concentrations were measured using the STA Compact Max® as percent activity, which is with respect to the normal coagulation factor concentration in a healthy person. A normal range for coagulation factor concentrations is typically 60–140% activity[57,58]. Plasma samples were removed from −80 °C storage and thawed at room temperature. Reagents were prepared with DiH$_2$O and left to stabilize for 30–60 min, as specified by the package insert. Owren-Koller diluent was used for patient samples, STA-Unicalibrator reagent was used to calibrate the system by measuring/defining ranges of new reagent lots (performed monthly), STA-System Control N+P and STA-Coag Control N+ABN were control reagents measured every 4 h and 8 h, respectively, and STA-Deficient reagent was used to measure the activity of a coagulation factor, e.g., STA-Deficient V was used for measuring factor V. The test automatically started after loading sample and reagents into the instrument. Given that quality control was repeated every four hours, coagulation factor concentration measurements were performed once for each sample.

### Thromboelastography

We obtained PPP by centrifuging each whole blood sample at 12,500 g for 10 min, and then at 15,000 g for another 15 min. Clot strength, functional fibrinogen, and platelet mapping were obtained using Thromboelastography, TEG 5000® (Haemonetics Corporation, Boston, Massachusetts, US). We followed the protocol for each assay in the user manual. At the beginning of each day, we checked the machine to ensure that it was on a flat level surface, and then performed quality control levels I and II. For the citrated native test, following blood sample collection, we pipetted 20 μL of CaCl$_2$, then pipetted 340 μL of whole blood into the bottom of the test cup. We placed the cup in the machine, locked the lever in the test position, and started the test from software.

### Model parameter fitting to experimental data

Model parameters were fit to experimental data using MATLAB SDO toolbox version 3.9.1. The input was defined as the CAT measurement for the plasma viscoelastic model, and as an impulse input with the desired magnitude, e.g., 5 pM of TF, in the case of the whole blood viscoelastic model. The output that was fit was individual TEG profile experimental data. Solver tolerance was set to $10^{-9}$. Starting from an initial parameter guess, the MATLAB SDO toolbox optimized parameter values of a transfer function model by minimizing the least square error between prediction and actual data using a trust region reflective algorithm. Following convergence, the finalized transfer function model parameters for each experimental sample were recorded.

## Data

This study was based on normal and trauma patient data arranged into 11 datasets as shown in Supplementary Fig. 1. Normal data was obtained from a set of blood samples from healthy individuals, with their CAT, TEG, and coagulation factor concentration measurements characterized according to standard laboratory protocols as explained above. These samples were purchased from commercial suppliers who are Institutional Review Board (IRB) exempt.

Dataset (7) consisted of ten normal 1 mL PPP samples (Precision BioLogic, Dartmouth, Nova Scotia, Canada). The plasma was not collected directly from donors but was instead obtained from FDA-regulated plasmapheresis centers for further manufacture into company products. Donors provided consent via a form that included the statement: "I, the undersigned, am donating blood to be used by [company] as they decide."

Dataset (8) consisted of five normal 10 mL whole blood samples (Innovative Research, Novi, Michigan, US) from which PPP was also obtained. Samples were obtained through Cedarlane Labs USA Inc., PO# 036950, Single Donor Human Whole Blood Na Citrate 10 mL, product code IWB1NAC10ml, from consented donors. Each participant provided consent before donation. All donors participated at will with full knowledge of donation usage. All donors were required to be over the age of 18 at the time of donation. Participant identity is kept confidential, and all samples were de-identified and assigned a number. Specifically, each donor was assigned an individual "donor number," which is not released to the public. Each time a sample was taken, the sample was given a unique "draw number" that was placed on the sample and stored for reference. Only the donors' age, sex, and race are provided. All donations were tested for the FDA-required viral markers.

Trauma patient data came from two studies. The Activation of Coagulation and Inflammation in Trauma study[59] was a single-center prospective cohort study that followed severely injured trauma patients from emergency department admission through discharge from hospitalization or death. Between February 2005 and May 2016, 1671 trauma patients (1367 male (81.45%), age 41.0 ± 18.6, ISS 17.7 ± 15.6) meeting criteria for highest triage activation level were enrolled into the study. Subsets of this dataset are used for various parts of this study, as indicated in the main text and Supplementary Fig. 1. Patient characteristics are in Supplementary Fig. 2. Exclusion criteria included patient age less than 15 years, pregnancy, incarceration, and transfer from outside hospital. This study was carried out with the approval of the University of California IRB (reference number 10-04417) under an IRB-approved delay and/or waiver of consent. Since it is not generally possible to consent a patient immediately after severe injury, and it is unethical even if possible due to the severely injured nature of the patients, a waiver of consent from the IRB allowed blood collection. Study coordinators then made multiple attempts to consent the patient or a surrogate. Written consent was obtained from enrolled patients or their families or, rarely, in certain circumstances where these could not be obtained, a waiver of consent was utilized for patients who could not consent due to their injuries and for whom a legally authorized representative could not be found despite documented attempts. Data were collected at admission, 6, 12, and 24 h after injury. Patient samples were kept and used if the patient or a surrogate consented, if the patient died, or if consent was unable to be obtained after multiple attempts to consent the patient or a surrogate. Samples and data were destroyed if patients or surrogates were contacted and refused consent.

The Control of Major Bleeding After Trauma (COMBAT) study[60], ClinicalTrials.gov number NCT01838863, was a single-center prospective randomized controlled trial on regulating hemorrhage after injury that evaluated the survival benefit of administering plasma prehospital during rapid ground transport to an urban Level I trauma center. Between April 2014 and March 2017, 144 trauma patients

were enrolled into the study, with as-treated analyses for 125 trauma patients (103 male (82.4%), age 36.5 ± 13.9, NISS 27.0 ± 19.4). Subsets of this dataset are used for various parts of this study, as indicated in the main text and Supplementary Fig. 1. Patient characterics are in Supplementary Fig. 2. Exclusion criteria included patient age less than 18 years, pregnancy, incarceration, and lack of consent. The study was conducted under Exemption from Informed Consent as part of a larger prospective clinical trial, and therefore exempted from needing written informed consent by the Colorado Multiple IRB (reference number 12-1349). Data were collected at admission, 2, 4, 6, 12, and 24 hours after injury.

### Reporting summary

Further information on research design is available in the Nature Portfolio Reporting Summary linked to this article.

## Data availability

Source data are provided with this article. Datasets are available at: https://github.com/SYBORGS-Lab/Viscoelastic-Clot-Model Source data are provided with this paper.

## Code availability

The MATLAB code and data it uses are available at: https://github.com/SYBORGS-Lab/Viscoelastic-Clot-Model. The patent pending algorithm code is available solely for noncommercial use. The code is subject to the following patent application number 63/498,624, copyright: © Copyright 2023 University of Florida Research Foundation, Inc. All commercial rights reserved by the University of Florida Research Foundation, Inc.

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

## Acknowledgements
M.J.C. was partially supported by NIH/NIGMS RM1 GM131968 and DHA Restoral Program FY2022. The authors are grateful to the research staff from the Cohen lab and the University of Colorado Denver for collecting patient data; and to Robert Martinez from the Menezes lab for assisting with TEG experimental setups.

## Author contributions
D.E.G. developed the models, conducted experiments on normal plasma and whole blood, performed data analysis, generated the figures, and drafted the article. A.J.V. confirmed model results, revised the figures, edited the article, and prepared model code and source data for dissemination. M.J.C. conceived the research, provided patient data, assisted with data analysis, and edited the article. A.A.M. conceived the research, guided model development, checked data analysis and model results, and revised and edited the article.

## Competing interests
The authors declare the following competing interests: a patent application has been filed that covers all material in this article.
