## [Peer Review File · Nature Communications]

Quick model-based viscoelastic clot strength predictions from blood protein concentrations for cybermedical coagulation controlReviewers' Comments:

Reviewer #1:

Remarks to the Author:

Thank you for the opportunity to review your manuscript. This is an elegant piece of work that builds nicely on the previous body of work by this group. This investigation tackles a very important real world clinical problem; the delay in obtaining the wealth of information available in TEG data. Going forward I would love to see the authors use this model to build on the work of Neilsen (reference #37) in more definitively characterizing the individual role of the all the coagulation factors roles in clot strength and dissolution.

I have one comment. On Page 21 Figure 8 you refer to the green, yellow, and red boxes in Figure 1 C; I believe you are referring to the boxes in Figure 1 B. You make a similar connection in the text (Page 23, line 386). In either case, It is confusing to the reader to use the same colors in Figure 1 B and 1C as it causes the reader to try to make a connection between clot composition and the process used to conduct the study.

Reviewer #2:

Remarks to the Author:

The authors developed a "systems" model to simulate clot formation and degradation following trauma.

This is an important problem. The authors have significantly advanced the field with this study. However, I have a few suggestions to improve the manuscript and promote open access to the findings:

- 1) What type of trauma - blunt force, penetrating, or burn was used for the patient data?
- 2) Please make the data and code available in a public version control repository, e.g., GitHub or GitLab. This will significantly advance the analysis of this important problem, leading to transparency and application in other clinical contexts.
- 3) Use open-source tools. MATLAB is not open source and has a restrictive, expensive license. Thus, the authors severely limit the transparency and utility of their model by using MATLAB. I understand MATLAB and its associated toolboxes are good at what they do, but MATLAB has a significant downside. Thus, I strongly suggest the authors make an open-source version of the model available in an open programming language, e.g., Python, Julia, C, etc. If not, someone in the community should (assuming #2).

Response to Reviewers

Nature Communications Manuscript NCOMMS-23-00139

Quick Model-Based Viscoelastic Clot Strength Predictions from Blood Protein Concentrations for Cybermedical Coagulation Control

Damon E. Ghetmiri, Alessia J. Venturi, Mitchell J. Cohen, and Amor A. Menezes

We thank Dr. Mackle for her efforts on behalf of our manuscript, and we are very grateful to the reviewers for their positive evaluation of our work. In this response to reviewers, the reviewer comments are in black, our **responses and descriptions of manuscript changes are in blue**, and our **highlights of new manuscript text are in red**. We have made a few small manuscript edits in response to reviewer comments to appropriately revise our previous submission.

Associate Editor

As detailed in the notes on requirements for resubmission, please make sure that your code is available for reviewer assessment. When resubmitting, you must provide a point-by-point response to the reviewers' comments. Please show all changes in the manuscript text file with track changes or colour highlighting. If you are unable to address specific reviewer requests or find any points invalid, please explain why in the point-by-point response.

Important: In addition to the above, you must comply with the following editorial requests; we will not be able to proceed with your revised manuscript otherwise. Please also see the *Nature Communications* formatting instructions, which you may find useful while preparing your revised manuscript.

We thank Dr. Mackle for obtaining reviews of our manuscript. Our revised submission addresses all of the editorial requests as well as the points that were raised by the reviewers. We also include our code and data in a single zip file for reviewer assessment. Per Reviewer 2's request, this code and data has been uploaded to a GitHub repository (<https://github.com/SYBORGS-Lab/Viscoelastic-Clot-Model>) that is currently set to private. This repository will be made public when our manuscript is published. We greatly appreciate the opportunity to submit a revised manuscript for another round of review.

Reviewer 1

Thank you for the opportunity to review your manuscript. This is an elegant piece of work that builds nicely on the previous body of work by this group. This investigation tackles a very important real world clinical problem; the delay in obtaining the wealth of information available in TEG data. Going forward I would love to see the authors use this model to build on the work of Neilsen (reference #37) in more definitively characterizing the individual role of the all the coagulation factors roles in clot strength and dissolution.

We are very grateful to the reviewer for their time spent evaluating our work, and for recognizing its merit. We enthusiastically agree with the reviewer about the immediate next step that this work suggests. We have thus added text that describes this next step in the Discussion section, as follows.

In parallel, our models can be refined by incorporating the effects of adding individual coagulation factor concentrations into samples of normal and trauma patient plasma and whole blood, to confirm model-predicted clotting outcomes and theoretically ground previous literature observations³⁷. Such work will also provide insight into treatment feasibility and efficacy from any proposed additional individual or combinatorial protein concentrations.

I have one comment. On Page 21 Figure 8 you refer to the green, yellow, and red boxes in Figure 1 C; I believe you are referring to the boxes in Figure 1 B. You make a similar connection in the text (Page 23, line 386). In either case, It is confusing to the reader to use the same colors in Figure 1 B and 1C as it causes the reader to try to make a connection between clot composition and the process used to conduct the study.

We truly appreciate the reviewer pointing out the confusion that arises from our color choices. Accordingly, we have revised the color palette of Fig. 1b to orange, cyan, and purple, so that there are no common colors with Fig. 1c, which remains blue, green, yellow, and red. We also now explicitly state the colors of Fig. 1b in the Fig. 1 caption, just as we had previously done with Fig. 1c, to clarify a color difference between panels.

Reviewer 2

The authors developed a “systems” model to simulate clot formation and degradation following trauma. This is an important problem. The authors have significantly advanced the field with this study. However, I have a few suggestions to improve the manuscript and promote open access to the findings.

We are very grateful to the reviewer for scrutinizing our manuscript, and for providing constructive comments that have enhanced our work and its accessibility.

(1) What type of trauma - blunt force, penetrating, or burn was used for the patient data?

We have added a summary of trauma patient characteristics for our utilized subsets of the ACIT and COMBAT datasets in Supplementary Fig. 2.

Dataset (6) Trauma Patient Characteristics

Characteristic	Mean \pm std. dev. or percentage (no. out of 97)
Age	36.3 \pm 13.6
Male / female	79.4% (77) / 20.6% (20)
NISS	27.9 \pm 19.7
Blunt / penetrating injury	51.5% (50) / 50.5% (49)
Alive / dead	88.7% (86) / 11.3% (11)

Dataset (10) Trauma Patient Characteristics

Characteristic	Mean \pm std. dev. or percentage (no. out of 24)
Age	49.5 \pm 23.0
Male / female	83.3% (20) / 16.7% (4)
ISS	27.5 \pm 15.5
Blunt / penetrating injury	91.7% (22) / 8.3% (2)
TBI not present / present	16.7% (4) / 83.3% (20)
Alive / dead	70.8% (17) / 29.2% (7)

Dataset (11) Trauma Patient Characteristics

Characteristic	Mean \pm std. dev. or percentage (no. out of 48)
Age	36.5 \pm 14.0
Male / female	79.2% (38) / 20.8% (10)
NISS	29.0 \pm 19.4
Blunt / penetrating injury	43.8% (21) / 58.3% (28)
Alive / dead	87.5% (42) / 12.5% (6)

Supplementary Figure 2: Summary of patient characteristics. Demographic and injury characteristics for Datasets (6), (10), and (11), Supplementary Fig. 1. The COMBAT dataset used the new injury severity score (NISS), while the ACIT dataset used the injury severity score (ISS). Three patients had both a penetrating and blunt injury across the three datasets, two in Dataset (6) and one in Dataset (11); these patients are counted in both categories.

(2) Please make the data and code available in a public version control repository, e.g., GitHub or GitLab. This will significantly advance the analysis of this important problem, leading to transparency and application in other clinical contexts.

(3) Use open-source tools. MATLAB is not open source and has a restrictive, expensive license. Thus, the authors severely limit the transparency and utility of their model by using MATLAB. I understand MATLAB and its associated toolboxes are good at what they do, but MATLAB has a significant downside. Thus, I strongly suggest the authors make an open-source version of the model available in an open programming language, e.g., Python, Julia, C, etc. If not, someone in the community should (assuming #2).

We wholeheartedly agree with the reviewer, especially because of our strong commitment to open access. Accordingly, we have uploaded our code and data to a GitHub repository (<https://github.com/SYBORGS-Lab/Viscoelastic-Clot-Model>) that is currently set to private. This repository will be made public when our manuscript is published. We also explored converting our MATLAB code to the open access MATLAB-like software Octave, but we found that we depended on MATLAB toolbox functionality that is as yet unavailable in Octave. We therefore enthusiastically endorse any community efforts to translate our code into Python, Julia, and/or C, given that our code and data will be public upon manuscript publication. In keeping with an editorial request, we have included our code and data in a single zip file for reviewer assessment.

Reviewers' Comments:

Reviewer #1:

Remarks to the Author:

Thank you for your comprehensive revisions.

Reviewer #2:

Remarks to the Author:

I thank the authors for responding to my previous issues, questions, and concerns. I look forward to the community effort to convert this codebase to PYTHON (or even better Julia).

Response to Reviewers

Nature Communications Manuscript NCOMMS-23-00139A

Quick Model-Based Viscoelastic Clot Strength Predictions from Blood Protein Concentrations for Cybermedical Coagulation Control

Damon E. Ghetmiri, Alessia J. Venturi, Mitchell J. Cohen, and Amor A. Menezes

We thank Dr. Mackle for her efforts on behalf of our manuscript, and we are very grateful to the reviewers for their positive evaluation of our work. In this response to reviewers, the reviewer comments are in black, our responses and descriptions of manuscript changes are in blue, and our highlights of new manuscript text are in red. We have made a few small manuscript edits in response to the received editorial requests to appropriately revise our previous submission.

Associate Editor

Your manuscript entitled “Quick Model-Based Viscoelastic Clot Strength Predictions from Blood Protein Concentrations for Cybermedical Coagulation Control” has now been seen again by our referees, whose comments appear below. In light of their advice I am delighted to say that we are happy, in principle, to publish a suitably revised version in *Nature Communications* under the open access CC BY license (Creative Commons Attribution 4.0 International License).

We therefore ask that you edit your manuscript to comply with our policies and formatting requirements and to maximise the accessibility and therefore the impact of your work.

Please see the attached document(s), listing a number of points that must be addressed. Failure to comply with our editorial requests will cause delays in accepting your manuscript. Please also see the *Nature Communications* formatting instructions for further information.

We thank Dr. Mackle for obtaining reviews of our manuscript. Our revised submission addresses all of the editorial requests as well as the points that were raised by the reviewers.

Reviewer 1

Thank you for your comprehensive revisions.

We are very grateful to the reviewer for their time spent evaluating our work, and for recognizing its merit.

Reviewer 2

I thank the authors for responding to my previous issues, questions, and concerns. I look forward to the community effort to convert this codebase to PYTHON (or even better Julia).

We are very grateful to the reviewer for scrutinizing our manuscript, and for providing constructive comments that have enhanced our work and its accessibility. We enthusiastically endorse community efforts to translate our code into Python, Julia, and/or C.